# Factors Associated with the Prolonged Use of Donor Human Milk at the Da Nang Hospital for Women and Children in Vietnam

**DOI:** 10.3390/nu16244402

**Published:** 2024-12-22

**Authors:** Hoang Thi Tran, Tuan Thanh Nguyen, Oanh Thi Xuan Nguyen, Le Thi Huynh, Roger Mathisen

**Affiliations:** 1Neonatal Unit and Human Milk Bank, Da Nang Hospital for Women and Children, Da Nang 50506, Vietnam; xuanoanh0901@gmail.com (O.T.X.N.); bongdiendien26@gmail.com (L.T.H.); 2Department of Pediatrics, School of Medicine and Pharmacy, The University of Da Nang, Da Nang 50206, Vietnam; 3Alive & Thrive, FHI 360 Global Nutrition, Hanoi 11022, Vietnam; rmathisen@fhi360.org

**Keywords:** breastfeeding, cesarean births, donor milk, human milk bank, low birthweight, infant, preterm births

## Abstract

Background and Objectives: Donor human milk (DHM) from a human milk bank (HMB) is used to feed low-birthweight (LBW) and preterm infants when mothers cannot provide their own breastmilk. The misuse of DHM could interfere with mothers’ breastmilk and weaken breastfeeding efforts. This study aimed to identify factors behind prolonged DHM usage during the first six years of Vietnam’s first HMB. Methods: Data were extracted from the Da Nang HMB’s digital monitoring system. We defined prolonged DHM use as four or more days in the neonatal unit and two or more days in postnatal wards. Results: Over six years, 25,420 infants received DHM, with 45.3% of the infants being female, 54.7% being male, 70.0% being born via cesarean section, and 77.2% being full-term. In the neonatal unit (*n* = 7001), 38.0% of infants used DHM for ≥4 days. Adjusted odds ratios (aORs) for prolonged use were 0.14 for infants weighing <1000 g, 0.78 for infants weighing 1000–<1500 g, and 0.67 for infants weighing ≥2000 g (*p* < 0.01), compared to those weighing 1500–<2000 g. Compared to gestational ages of 32–<34 weeks, the aORs were 0.26 for <28 weeks, 0.71 for 34–<37 weeks, and 0.35 for ≥37 weeks (*p* < 0.01). In postnatal wards (*n* = 18,419), 53.1% of infants used DHM for ≥2 days. Compared to term, normal-weight infants, the aORs were 1.25 for LBW–preterm, 1.17 for LBW–term, and 1.21 for normal-weight–preterm infants (*p* < 0.05). Prolonged DHM use was associated with cesarean births in neonatal units (aOR 2.24, *p* < 0.01) and postnatal wards (aOR 1.44, *p* < 0.01). Conclusions: DHM is used briefly to bridge nutritional gaps and transition to mothers’ breastmilk, but LBW, preterm births, and cesarean births are linked to prolonged use. Healthcare providers should support those at risk of prolonged DHM use and prioritize reducing unnecessary cesarean births.

## 1. Introduction

Breastfeeding and human milk are the normative standards for infant and young child feeding [1]. Breastmilk is a special bioactive fluid that adapts its composition to meet the nutritional, developmental, and immune needs of an infant and young child. While breastmilk and lactation function as a biological system, breastfeeding itself is a behavior influenced by various social and cultural norms, as well as the characteristics of both the mother and the infant and young child [2]. When a mother’s own milk is not available, donor human milk (DHM) from a human milk bank (HMB) is the alternative recommended by the World Health Organization (WHO) for low-birthweight (LBW) infants [3,4]. HMBs recruit donors; collect, store, screen, and pasteurize raw breastmilk; and administer DHM to infants in need. To date, there are more than 700 HMBs globally [5].

The use of DHM has resulted in decreases in the incidence of necrotizing enterocolitis [6], the prevalence of sepsis [7], and the incidence of bronchopulmonary dysplasia [8] in comparison to preterm infants who were supplemented with preterm formula, which helps reduce the duration of hospital stays and treatment costs [9,10]. In some HMBs, DHM has been used to bridge the infant feeding and nutrition gap when mothers’ own milk is not yet available or sufficient [11]. Evidence shows that DHM prevented the use of infant formula, increased breastfeeding and exclusive breastfeeding practices during the hospital stay and after the hospital discharge [12,13], and prevented wet nursing, which might pose a risk of transmission of diseases to small and vulnerable infants [14]. An HMB is not just a place for collecting, storing, processing, and distributing DHM; it should also promote breastfeeding and support mothers in overcoming breastfeeding difficulties [15].

However, the potential misuse of DHM includes providing it to infants for an extended period, which can affect the mother–infant dyad’s breastfeeding capacity and practices due to less direct breastfeeding to stimulate breastmilk production [11]. It might also weaken efforts to support breastfeeding within the hospital [11]. Not directly breastfeeding can lead to earlier weaning, reduced suckling from the breast, and, later, linguistic development issues for the child [16,17]. There is also a concern relating to the financial cost and environmental impact related to the collection, processing, storage, and use of DHM [18,19]. Therefore, it is necessary to examine the factors associated with the prolonged use of DHM to develop evidence-based interventions [20].

The first HMB in Vietnam was established at the Da Nang Hospital for Women and Children in 2017 [21]. The HMB provides DHM to small vulnerable infants, but also some healthy infants, when their mothers’ own milk supplies are unavailable or compromised. For decades, this hospital has been a WHO Center of Excellence for early essential newborn care (EENC) and kangaroo mother care (KMC) and has a strong foundation for breastfeeding promotion, protection, and support [21]. In 2020, a data review suggested the potential overuse of DHM when it was not necessary or for longer than needed. The hospital implemented several strategies to reduce dependence on DHM, such as continuing to strengthen the implementation of immediate and prolonged EENC and KMC, as well as consultation and support for breastfeeding from the prenatal to the postnatal period. The indication for DHM (need, volume, and duration) was assessed by medical doctors at least once a day by evaluating nutritional needs (e.g., number of days from birth, weight, weight gain) and the availability of the mother’s own milk, as well as clinical symptoms and biochemical test results. Activities such as auditing indications and capacity building for hospital staff were also conducted. The intervention was based on the Decision Tree for Donor Human Milk, which was designed to help guide the prioritization and allocation of donor human milk, ensuring optimal nutrition and preventing misuse [22]. Stricter indications and close monitoring of DHM use would help reduce the volume and duration of DHM use, but what is the impact of these interventions?

To address these literature gaps, this study was conducted to investigate factors related to the prolonged use of DHM among these infants in neonatal units and postnatal wards.

## 2. Methods

### 2.1. Study Setting

The Da Nang Hospital for Women and Children is a tertiary hospital for obstetrics, gynecology, and pediatrics. It has a capacity of 1200 beds, serving Da Nang and other provinces in Central South Vietnam. There are approximately 15,000 births in this hospital annually. This level 4 neonatal unit has a capacity of 100 beds and admits around 4000 infants per year, with more than 30% being preterm [23]. Term and healthy newborns, as well as those with mild prematurity and low birthweight who are otherwise healthy and feeding well, are typically assigned to postnatal wards.

The criteria for recipients of DHM from the HMB include small and vulnerable infants (i.e., preterm or sick newborns), abandoned infants, and healthy infants who do not have adequate supplies of their mothers’ own milk [24]. Because this HMB has surplus DHM, the main indication was for newborns whose mothers’ own milk is not sufficient [11].

### 2.2. Study Data

Data were routinely collected during the operation of the HMB in the first six years, from 1 February 2017 to 31 January 2023. The data were collected and updated daily to a locally developed, web-based digital monitoring system that captures certain characteristics of the recipients, including the age, mode of birth, and DHM volume use [11]. There are no differences in the variables over the analysis period. For this study, we extracted data directly from the HMB monitoring system and removed identifiable information from all records before data analysis. For mothers, we kept only the province of residence and age while deleting the name, address, phone number, ID card number, and health insurance number. For infants, we excluded their names, date of birth, and admission date.

### 2.3. Variables

The dependent outcome variable for this study was the duration of DHM use, which was the summation of each recipient’s DHM use duration. The duration of DHM use was calculated by subtracting the first DHM use date from the date of the last use. The average duration per recipient was calculated using the median duration of DHM per recipient. For the regression analysis, we dichotomized the total duration using the median value; prolonged use was defined as equal to or greater than the median duration per recipient of 4 days in the NICU and 2 days in the postnatal ward. Of note, this hospital is formula-free, which means that this is the duration before all infants successfully transitioned to exclusive breastmilk from their mothers.

The main exposure variable is the year of birth before 2020 vs. during and after 2020. We used this cut-off point for several reasons. First, 2020 is the middle of the 6-year period, so we wanted to test whether the use of DHM had increased over time. Second, 2020 is the peak of the COVID-19 pandemic, which led to changes in hospital procedures that could have affected the donation and use of DHM. Since 2020, the hospital has implemented strategies to minimize unnecessary DHM use, shorten its duration, and strengthen breastfeeding support.

Maternal residency includes the Da Nang municipality and other provinces; the recipient place includes the neonatal unit and the postnatal ward. Neonatal characteristics include sex, mode of birth, gestational age at birth, and birth weight. The amount of DHM use was recorded as daily and total amounts.

### 2.4. Data Analysis

We first performed a descriptive analysis to review the trends in the amount and duration of DHM used by the location of the recipient within the hospital: neonatal unit and postnatal ward. We presented descriptive statistics as the mean and standard deviation for normally distributed continuous variables or the median and interquartile range for skewed continuous variables. Counts and percentages were used for binary and categorical variables.

Secondly, we performed a crude binary analysis to examine the association among neonatal characteristics, maternal residency, and birth before the year 2020 for the prediction of the prolonged use of DHM (≥4 days for infants at neonatal units and ≥2 days in postnatal wards) using the chi-squared test.

Thirdly, we performed adjusted logistic regression using Wald statistics with all the abovementioned covariates. Four adjusted logistic regression models were developed. The first model controlled for birth weight and gestational age in a continuous form. The second model controlled for birth weight in a categorized form and gestational age in a continuous form. The third model controlled for birth weight in a continuous form and gestational age in a categorical form. The fourth model, for infants in postnatal wards, controlled for a four-category variable defined by birth weight status and gestational age. The results were presented as crude and adjusted odds ratios with 95% confidence intervals. A two-tailed *p*-value of <0.05 defined statistical significance. We used IBM SPSS Statistics (version 26) to analyze the data.

## 3. Results

From 1 February 2017 to 31 January 2023, 25,420 infants received DHM from the Da Nang HMB, with 45.3% being female and 54.7% being male. Most recipients were born via cesarean section (70.0%) and were full-term (77.2%). These characteristics remained relatively consistent over time (Table 1). The prevalence of infants receiving DHM during their treatment in neonatal units ranged from approximately 900 to 1450 each year, averaging 27.5%. This prevalence decreased from 37.0% in the initial years to 20.1% in the sixth year (Table 1).

In the neonatal units, almost all infants who received DHM began receiving it on the first day of life, with an average duration of about 3–4 days (median: 3 days, IQR: 2–4 days; mean: 4.0 days, SD: 4.6 days). There was a trend toward shorter DHM use across birth weight and gestational age categories in 2020–2023 compared to 2017–2019 (Table 2). Among infants treated in neonatal units, 62.0% received DHM for less than 4 days, and only 2.8% required DHM for 14 days or more (Figure 1). There was a general trend toward shorter DHM use in neonatal units (Figure 1).

In postnatal wards, almost all infants who received DHM began receiving it on the first day of life, with an average duration of about 1–2 days (median: 2 days, IQR: 1–2 days; mean: 1.8 days, SD: 0.9 days). There was a trend toward shorter DHM use in 2019–2023 compared to 2017–2018 (Table 2). In postnatal wards, 46.9% of infants used DHM for less than 2 days, and 48.6% used DHM for 2 to less than 4 days (Figure 1). There was an increasing trend toward shorter DHM use, especially under 2 days, in postnatal wards (Figure 1).

Regarding the total per capita volume of DHM in neonatal units, the median volume of DHM used was 360 mL (IQR: 186–633 mL). Notably, the group of infants with 32–<34 weeks of gestation required the largest amount (375 mL), while the extremely preterm group (<28 weeks of gestation) required the smallest amount (20 mL) (Table 3). The extremely preterm group also required the shortest duration of DHM use compared to other groups. In postnatal wards, the median volume of DHM used was 150 mL (IQR: 80–200 mL) and did not vary by year (Table 3).

Over the six years, among the 7001 infants receiving DHM in neonatal units, 38.0% received DHM for 4 days or more (prolonged use) (Figure 1). In the crude estimation, the prolonged use of DHM in neonatal units showed an association with birth weight categories (highest in 1500–<1500 g) and gestational age categories (highest in 23–<34 weeks) (Table 4). In the adjusted model (Table 4, Figure 2), an association was retained for birth weight; compared to infants weighing 1500–<2000 g, the aORs were 0.14 (95% CI: 0.10–0.20), 0.78 (95% CI: 0.65–0.94), and 0.67 (95% CI: 0.58–0.77) for those with birth weights of <1000 g, 1000–<1500 g, and ≥2000 g, respectively (Model 2). There is also a clear association with gestational age; compared to infants with gestational ages of 32–<34 weeks, the aORs were 0.26 (95% CI: 0.18–0.36), 0.97 (95% CI: 0.81–1.16), 0.71 (95% CI: 0.61–0.83), and 0.35 (95% CI: 0.29–0.43) for those with gestational ages of <28 weeks, 28–<32 weeks, 34–<37 weeks, and ≥37 weeks, respectively (Model 3, Table 4, Figure 2). All models show that cesarean births and births before 2020 were positively associated with the prolonged use of DHM (Table 4).

Among the 18,419 infants receiving DHM in postnatal wards, 53.1% received DHM for 2 days or more (prolonged use) (Figure 1). Table 5 shows that prolonged DHM use was higher in low-birthweight (aORs of 1.12 (95% CI: 0.99–1.27) (Model 2) and preterm infants (aORs of 1.29, (95% CI: 1.15–1.45) (Model 3). In Model 4 (Table 5, Figure 2), compared to those with normal weight and term infants (≥2500 g and ≥37 weeks), the aORs were 1.25 (95% CI: 1.09–1.43), 1.17 (95% CI: 1.00–1.36), and 1.21 (95% CI: 1.04–1.40) for infants who were LBW–preterm (<2500 g and <37 weeks), LBW–term (<2500 g and ≥37 weeks), and normal weight–preterm (≥2500 g and <37 weeks), respectively. Regardless of the models, mothers from Da Nang, cesarean births, and births before 2020 were positively associated with the prolonged use of DHM for infants in postnatal wards (Table 5).

## 4. Discussion

### 4.1. Duration of DHM Use in the Da Nang HMB Is Shorter than in Other HMBs

The duration of DHM use in the Da Nang HMB was shorter than that reported by other HMBs worldwide. On average, DHM use in Da Nang lasted only 4 days. In comparison, during six years of operation (2005–2011) in Taiwan, 551 recipients of DHM were documented, and 40% of them required DHM for more than one month [25]. Similarly, the first HMB in East China supplied a total of 5775 L of DHM to 9207 infants over an 8-year period. Among these recipients, most were preterm (83.3%), with 41.7% having a gestational age of 34–37 weeks. While 94.3% received DHM for less than 15 days, the average duration of use was 4.5 days [26]. The DHM use in neonatal units of the Da Nang Hospital was also shorter than that reported in Scotland and Poland [25,27,28]. A study conducted in Chicago, USA (2013–2015), involving 169 very LBW infants, found that only 46.4% had exclusive breastmilk by day 14 despite using both mothers’ own milk and DHM [29].

LBW and preterm births were associated with prolonged DHM use. Preterm infants, in particular, face challenges such as immature neural development and respiratory problems, which hinder their ability to coordinate suckling, swallowing, and breathing [30]. Additionally, the separation of preterm infants from their mothers in the first few days due to neonatal or maternal complications can delay the initiation of breastfeeding with their mothers’ own milk [31]. However, early and prolonged skin-to-skin contact (SSC), KMC, and timely oral feeding introduction can help overcome these barriers. Keeping mothers and preterm infants together in neonatal units allows mothers to observe and respond to their infant’s feeding cues, breastfeed more frequently, and foster bonding and engagement, which helps establish and maintain breastfeeding [32]. Rooming-in improves sleep for both mothers and babies, boosts mothers’ confidence, stabilizes babies’ body temperatures and blood sugar levels, and reduces stress, leading to less crying and more contentment [32,33,34]. Studies have shown that co-locating mothers and infants in neonatal units improves breastfeeding practices and reduces the length of hospital stays [33,35].

At the Da Nang Hospital for Women and Children, SSC, KMC, and breastfeeding have been actively promoted since 2014 [31]. Preterm infants, even those on respiratory support, begin KMC immediately after birth if they are stable on continuous positive airway pressure (CPAP) with less than 30% oxygen requirements. Transporting preterm infants to neonatal units while maintaining SSC and continuing KMC is a key goal. Despite limited space and facilities for family-centered care, parents at the Da Nang Hospital have shown a remarkable commitment to practicing KMC, even for extended periods exceeding three months for infants born at less than 28 weeks of gestation [31].

These efforts have resulted in significant improvements in breastfeeding outcomes. Preterm infants receive their mothers’ own milk earlier and achieve full breastfeeding more quickly, leading to a much shorter duration of DHM use. For instance, a report from Italy (2017–2021) involving 97 very LBW infants indicated that each infant required, on average, 2900 mL of DHM and transitioned to their mothers’ own milk by day 4 of life [36]. By contrast, preterm infants in the Da Nang Hospital neonatal units required only 360 mL of DHM before transitioning to their mothers’ own milk. These findings highlight that the Da Nang HMB has optimized the use of DHM to bridge the gap in nutritional needs and transition to using mothers’ own milk and direct breastfeeding.

### 4.2. Association Among Birth Weight, Gestational Age, and Prolonged Use of DHM

In neonatal units, the use of DHM demonstrated an association with the birth weight and gestational age categories. Very LBW or extremely preterm infants often required only a small amount of breastmilk, which their mothers could typically provide for a shorter duration. However, it is important to note that we were unable to link our data with mortality outcomes. It is likely that higher mortality rates among these critically ill infants during their hospital stays contributed to a shorter duration of DHM use, as DHM would no longer be needed at the time of death [37,38]. Conversely, heavier or more mature infants required larger quantities of breastmilk, which may have exceeded their mothers’ supply, leading to prolonged reliance on DHM.

The HMB in Da Nang also provides DHM to infants in postnatal wards during the first few days after birth, bridging the gap when the mother’s milk is not yet available or sufficient [4]. In this study, term and normal-weight infants typically used DHM for shorter durations. Preterm and LBW infants faced additional challenges in accessing their mothers’ own milk [39,40]. These infants often cannot breastfeed directly due to poor latching and weak suckling reflexes, which delay the milk let-down reflex in mothers. Preterm infants have a higher risk of postnatal resuscitation or respiratory distress, interrupting the instinctive steps of infant feeding and complicating latching efforts. This disruption, combined with maternal factors such as stress, inexperience, and improper breastfeeding techniques, can delay or reduce breastmilk production [39,40]. First-time mothers may struggle to hold or breastfeed their infants correctly. Women not receiving the support they need to practice breastfeeding, including skills for holding the baby for a good latch, can lead to insufficient breastmilk intake by the infant, excessive crying, and maternal stress, perpetuating the perception of inadequate breastmilk supply and further delaying breastmilk secretion [39,40].

While LBW and low gestational age remain significant barriers to breastfeeding, these challenges can be mitigated through adherence to the *Ten Steps for Successful Breastfeeding* [34], particularly by preventing mother–child separation [34], along with strategies to address breastfeeding challenges [39,40].

### 4.3. Cesarean Sections as a Barrier to Breastfeeding and DHM Use

Cesarean sections were associated with prolonged DHM use in the Da Nang Hospital. Previous studies have shown that infants born via cesarean sections required longer durations of DHM compared to those born vaginally [13,41,42]. Evidence indicates that cesarean sections delay the initiation of breastfeeding, reduce breastmilk production, and exacerbate maternal pain and discomfort, all of which negatively impact breastfeeding practices [43,44]. A population-based study in Vietnam similarly found that cesarean section births were associated with late initiation of breastfeeding, increased use of commercial formula in the first three days of life, and subsequent early cessation of breastfeeding [45]. Additionally, the prevalence of cesarean births among infants receiving DHM was 70.0%, which is higher than the hospital average of 53.9% to 60.6% between 2017 and 2022. This observation suggests that cesarean births may also increase the use of DHM. However, our data did not allow for this analysis.

Globally, cesarean rates are rising despite the WHO recommendation to limit cesarean sections to medically indicated cases, with a target rate of less than 15% [46]. In Vietnam and other countries in the Western Pacific region, the rising rate of cesarean sections is driven by multifactorial reasons. At the Da Nang Hospital for Women and Children, all mothers and breathing infants receive SSC for at least 90 min immediately after birth, along with intensive support for breastfeeding. Vietnam has national guidelines on EENC for both vaginal and cesarean births [47,48]. However, cesarean sections remain a significant barrier to early breastfeeding initiation, exclusive breastfeeding during hospital stay, and subsequent breastfeeding practices [45], which contributes to prolonged DHM use. Thus, strategies to reduce cesarean rates should be prioritized in healthcare settings [46]. These efforts are critical to improving breastfeeding outcomes, reducing dependency on DHM, and promoting maternal and neonatal health.

### 4.4. Shortened Duration of DHM Use After 2020 in Neonatal Units and Postnatal Wards

The period before 2020 significantly influenced prolonged DHM use. We chose 2020 as a cut-off point to analyze trends in DHM usage, coinciding with the sixth year of the HMB’s operation and the onset of the COVID-19 pandemic. Findings revealed a correlation between the pre-2020 period and prolonged DHM usage, suggesting positive shifts in HMB practices over time. Notably, the presence of the Da Nang HMB has not led to dependency on DHM but has actively contributed to enabling breastfeeding [49].

During the COVID-19 period, several measures were implemented to prevent transmission, including limiting visits from relatives and isolating infected individuals, which may have reduced breastfeeding [50]. Despite some countries, including Vietnam, not following WHO guidelines [51], the Da Nang Hospital for Women and Children adhered to them. As a result, unlike many other locations where the pandemic disrupted breastfeeding rates and HMB operation, the Da Nang HMB maintained its services, including milk donation and distribution, without interruption. This resilience highlights the strength and sustainability of the HMB and its effectiveness in supporting breastfeeding during challenging times. During and after the COVID-19 pandemic, the duration of DHM use remained short due to consistent policies emphasizing SSC and breastfeeding, even for mothers with COVID-19 [50].

In 2020, the trends in DHM usage at the Da Nang HMB were analyzed [49]. Recognizing the need to reduce dependence on DHM, the hospital implemented guidelines for prescribing DHM and monitoring the availability of mothers’ own milk. Increased breastfeeding support efforts also improved the volume and availability of mothers’ own milk. This includes medical doctors assessing the need, volume, and duration of DHM daily by evaluating nutritional needs (e.g., days from birth, weight, weight gain), the availability of mothers’ milk, clinical symptoms, and biochemical test results. Auditing indications and capacity building for hospital staff were also conducted. The intervention was based on the Decision Tree for Donor Human Milk, which was designed to help guide the prioritization and allocation of donor human milk, ensuring optimal nutrition and preventing misuse [22]. As a result, DHM usage duration was shortened in neonatal and postnatal wards.

Interestingly, the duration of DHM use in postnatal wards was shorter for infants of mothers from other provinces, but this trend did not extend to neonatal units. In neonatal units, discharge decisions are typically standardized and based on the infant’s clinical condition. However, in postnatal wards, discharge timing may be flexible and influenced by the mother’s health and family convenience and preference. Mothers from other provinces may opt for earlier discharge to return home. Additionally, the out-of-pocket costs for DHM might be more manageable for Da Nang municipality residents than those from other provinces.

### 4.5. Strengths and Limitations

Utilizing secondary data from a digital monitoring system proved to be cost-effective and minimized recall bias. The system’s structured forms, pre-coded options, and built-in verification functions significantly reduced the workload for healthcare workers and ensured high data quality. Regularly using this monitoring system also optimized HMB operation, enforced adherence to standardized protocols, and facilitated networking and information sharing among HMBs.

However, the reliance on secondary data limited the ability to explore other factors influencing DHM use, necessitating targeted data collection and analysis for a more comprehensive understanding. The electronic data from this HMB were incompatible with other data from medical records, including information on clinical symptoms, laboratory test results, management, clinical progression, the duration of hospital stays, neonatal outcome at discharge, and records from follow-up visits. These hospital records were typically paper-based (before 2021) and would require significant effort to digitize and link with the HMB digital data. Despite these limitations, this study provides valuable insights into the determinants of prolonged DHM use, which can aid in optimizing care and resource allocation in neonatal and postnatal wards. Additionally, the findings underscore the importance of interventions such as reducing unnecessary cesarean section births.

## 5. Conclusions

In conclusion, in this hospital, DHM has been used for a short duration to bridge the gap in nutritional needs and transition to the use of mothers’ own milk and direct breastfeeding. This makes the use of DHM a viable solution for bridging the nutritional gaps in infants when their mothers’ own milk is not yet available or sufficient, thereby contributing to recommended breastfeeding practices during the hospital stay and over time. Yet, this study identified several factors associated with prolonged use of DHM, including LBW, preterm births, cesarean births, and births before 2020. These findings highlight the need for targeted interventions to optimize the use of DHM and transition back to mothers’ own milk and direct breastfeeding. It is recommended that healthcare providers do not promote DHM as a substitute for breastmilk and breastfeeding and instead provide specialized support and counseling for those at risk for prolonged DHM use. Additionally, strategies to reduce unnecessary cesarean births should be prioritized to enhance breastfeeding outcomes.

## Figures and Tables

**Figure 1 nutrients-16-04402-f001:**
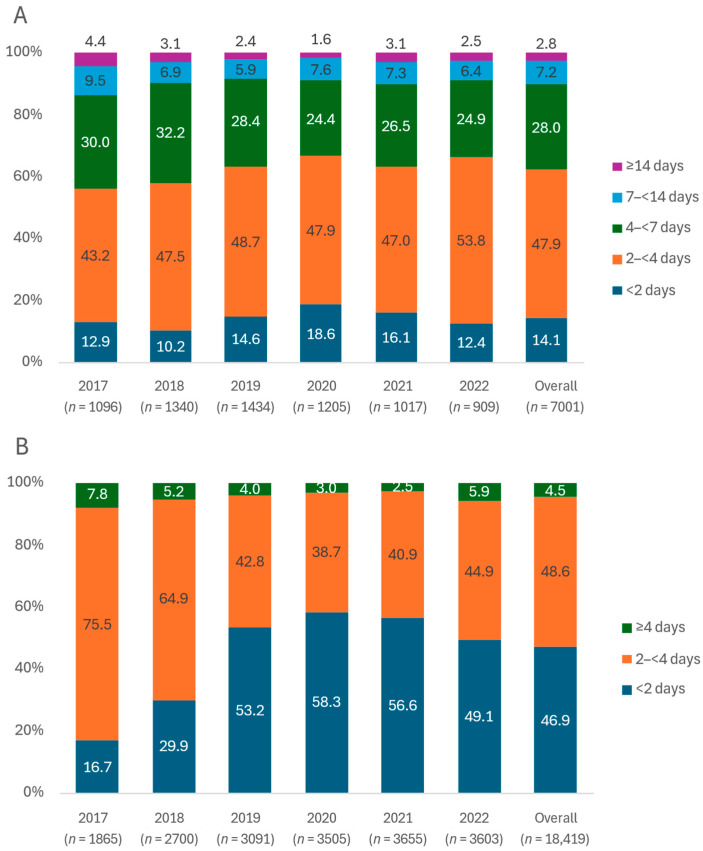
Distribution (%) of duration of use of DHM (days) in neonatal units (**A**) and postnatal wards (**B**) by year.

**Figure 2 nutrients-16-04402-f002:**
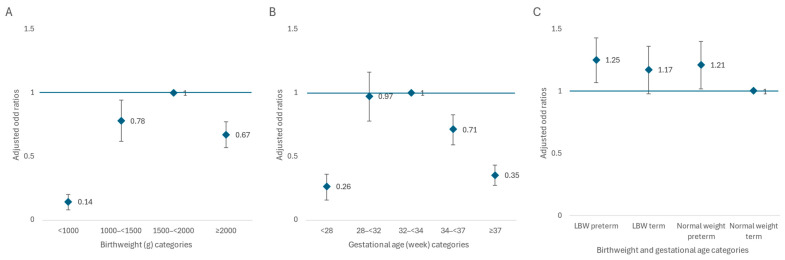
Adjusted ORs of prolonged DHM use in neonatal unit for birth weight categories (**A**) and gestation age (**B**) and postnatal wards for birth weight and gestational age categories (**C**). All models controlled for child sex, mothers from other provinces, cesarean births, and births before 2020; the model in panel A controlled for gestational age in a continuous form; and the model in panel A controlled for birth weight in a continuous form.

**Table 1 nutrients-16-04402-t001:** Characteristics of infants receiving DHM at the Da Nang Hospital for Women and Children by year.

	2017	2018	2019	2020	2021	2022	Overall
	(*n* = 2961)	(*n* = 4040)	(*n* = 4525)	(*n* = 4710)	(*n* = 4672)	(*n* = 4512)	(*n* = 25,420)
Sex, male, *n* (%)	1680 (56.7)	2234 (55.3)	2544 (56.2)	2514 (53.4)	2521 (54.0)	2407 (53.3)	13,900 (54.7)
Cesarean births, *n* (%)	2072 (70.0)	2781 (68.8)	3179 (70.3)	3405 (72.3)	3167 (67.8)	3177 (70.4)	17,799 (70.0)
Gestational age at birth (week):							
Average, mean (SD)	37.2 ± 3.0	37.3 ± 2.8	37.3 ± 2.8	37.6 ± 2.5	37.6 ± 2.6	37.6 ± 2.5	37.4 ± 2.7
<28, *n* (%)	32 (1.1)	42 (1.0)	47 (1.0)	29 (0.6)	60 (1.3)	55 (1.2)	265 (1.0)
28–<32, *n* (%)	151 (5.1)	160 (4.0)	180 (4.0)	147 (3.1)	163 (3.5)	96 (2.1)	897 (3.5)
32–<34, *n* (%)	222 (7.5)	240 (5.9)	274 (6.1)	215 (4.6)	170 (3.6)	169 (3.7)	1290 (5.1)
34–<37, *n* (%)	430 (14.5)	641 (15.9)	675 (14.9)	598 (12.7)	461 (9.9)	532 (11.8)	3337 (13.1)
Full-term (≥37 weeks)	2126 (71.8)	2957 (73.2)	3349 (74.0)	3721 (79.0)	3818 (81.7)	3660 (81.1)	19,631 (77.2)
Birth weight (g):							
Average, mean (SD)	2826 ±756	2876 ±727	2880 ±717	2969 ±684	2964 ±675	2940 ±655	2916 ±701
<1000, *n* (%)	40 (1.4)	43 (1.1)	56 (1.2)	41 (0.9)	64 (1.4)	51 (1.1)	295 (1.2)
1000–<1500, *n* (%)	156 (5.3)	145 (3.6)	155 (3.4)	133 (2.8)	140 (3.0)	111 (2.5)	840 (3.3)
1500–<2500, *n* (%)	627 (21.2)	861 (21.3)	907 (20.0)	728 (15.5)	626 (13.4)	698 (15.5)	4447 (17.5)
≥2500, *n* (%)	2138 (72.2)	2991 (74.0)	3407 (75.3)	3808 (80.8)	3842 (82.2)	3652 (80.9)	19,838 (78.0)
Under treatment in neonatal units, *n* (%)	1096 (37.0)	1340 (33.2)	1434 (31.7)	1205 (25.6)	1017 (21.8)	909 (20.1)	7001 (27.5)
Mothers from other provinces, *n* (%)	1499 (50.6)	2157 (53.4)	2432 (53.7)	2118 (45.0)	1733 (37.1)	2114 (46.9)	12,053 (47.4)

**Table 2 nutrients-16-04402-t002:** Number of days of DHM use by infants in neonatal units and postnatal wards.

	2017	2018	2019	2020	2021	2022	Overall
**Neonatal units**	(*n* = 1096)	(*n* = 1340)	(*n* = 1434)	(*n* = 1205)	(*n* = 1017)	(*n* = 909)	(*n* = 7001)
Day of age started using, median (IQR)	1 (1–1)	1 (1–1)	1 (1–1)	1 (1–1)	1 (1–2)	1 (1–1)	1 (1–1)
Day of age stopped using, median (IQR)	5 (4–6)	5 (4–6)	4 (3–6)	4 (3–5)	4 (3–6)	4 (3–5)	4 (3–6)
Mean duration, mean (SD)	4.5 ± 5.5	4.2 ± 5.0	3.8 ± 4.4	3.5 ± 3.1	4.0 ± 4.9	3.8 ± 4.2	4.0 ± 4.6
Median duration of using (days), median (IQR; min-max)	3 (2–5; 1–60)	3 (2–4; 1–79)	3 (2–4; 1–70)	3 (2–4; 1–41)	3 (2–4; 1–63)	3 (2–4; 1–67)	3 (2–4; 1–79)
By birth weight (g):							
<1000	3 (1–5)	3 (2–4)	2 (2–3)	2 (1–3)	3 (2–3)	2 (1–3)	3 (2–3)
1000–<1500	4 (3–7)	4 (3–5)	4 (3–5)	3 (3–5)	4 (3–5)	3 (3–4)	4 (3–5)
1500–<2000	4 (3–5)	4 (3–5)	4 (3–5)	4 (3–5)	4 (3–6)	3 (2–4)	4 (3–5)
≥2000	3 (2–4)	3 (2–4)	3 (2–4)	2 (2–4)	3 (2–4)	3 (2–4)	3 (2–4)
By gestational age (week):							
<28	3 (1–5)	3 (2–4)	2 (2–3)	2 (2–3)	3 (2–4)	2 (2–3)	2 (2–3)
28–<32	4 (3–7)	4 (3–5)	3 (3–5)	4 (3–5)	4 (3–5)	3 (2–4)	4 (3–5)
32–<34	4 (3–5)	4 (3–5)	4 (3–5)	3 (3–5)	4 (3–6)	3 (2–4)	4 (3–5)
34–<37	4 (3–5)	4 (3–5)	4 (3–5)	3 (3–5)	3 (2–5)	3 (2–4)	3 (3–5)
**Postnatal wards**	(*n* = 1865)	(*n* = 2700)	(*n* = 3091)	(*n* = 3505)	(*n* = 3655)	(*n* = 3603)	(*n* = 18,419)
Day of age started, median (IQR)	2 (1–2)	1 (1–2)	1 (1–2)	1 (1–2)	1 (1–2)	1 (1–2)	1 (1–2)
Day of age stopped, median (IQR)	4 (3–4)	3 (3–4)	3 (3–4)	3 (2–4)	3 (2–4)	3 (3–4)	3 (3–4)
Mean duration, mean (SD)	2.2 ± 0.9	2.0 ± 0.9	1.7 ± 1.0	1.6 ± 0.8	1.6 ± 0.8	1.8 ± 1.0	1.8 ± 0.9
Median duration of using (days), median (IQR; min-max)	2 (2–2; 1–8)	2 (1–2; 1–8)	1 (1–2; 1–14)	1 (1–2; 1–8)	1 (1–2; 1–7)	2 (1–2; 1–11)	2 (1–2; 1–14)

**Table 3 nutrients-16-04402-t003:** Median (IQR) volume (mL) of total DHM used by infants during their stay in neonatal units and postnatal wards.

	2017	2018	2019	2020	2021	2022	Overall
**Neonatal units**	(*n* = 1096)	(*n* = 1340)	(*n* = 1434)	(*n* = 1205)	(*n* = 1017)	(*n* = 909)	(*n* = 7001)
Overall (mL)	342 (164–652)	409 (232–690)	371 (216–624)	341 (168–592)	330 (151–645)	331 (164–574)	360 (186–633)
By birth weight (g):							
<1000	28 (12–167)	29 (14–91)	16.5 (11–41)	18 (6–66)	23 (11–47)	14 (10–27)	21 (11–48)
1000–<1500	209 (78–527)	216 (108–475)	248 (124–452)	165 (90–318)	204 (71–352)	138 (60–318)	198 (90–432)
1500–<2000	352 (196–662)	411 (259–701)	380 (242–583)	388 (240–646)	346 (205–730)	307 (194–456)	366 (228–625)
≥2000	396 (200–670)	445 (288–731)	405 (252–691)	374 (200–617)	382 (208–710)	418 (240–678)	404 (230–684)
By gestational age (week)							
<28	33 (9–158)	27 (14–57)	13 (10–43)	18 (9.5–78)	24 (11–49)	15 (10–36)	20 (11–49)
28–<32	200 (69–652)	251 (128–471)	255 (126–460)	204 (89–478)	219 (71–399)	129 (60–275)	219 (90–450.8)
32–<34	372 (206–615)	424 (248–701)	409 (255–628)	376 (200–625)	404 (203–809)	304 (194–462)	375 (222–635)
34–<37	366 (288–625)	366 (193–658)	345 (185–588)	312 (140–553)	274 (96–579)	237 (98–428)	310 (141–585)
**Postnatal wards**	(*n* = 1865)	(*n* = 2700)	(*n* = 3091)	(*n* = 3505)	(*n* = 3655)	(*n* = 3603)	(*n* = 18,419)
Overall (mL)	180 (140–250)	140 (80–200)	120 (80–200)	120 (80–200)	120 (80–200)	160 (80–240)	150 (80–200)

**Table 4 nutrients-16-04402-t004:** Factors associated with a prolonged duration of DHM use in neonatal units (*n* = 7100).

	Duration of Using DHM		Binary Analysis	Adjusted Model 1	Adjusted Model 2	Adjusted Model 3
	Less than 4 days (*n* = 4342)	4 days or more (*n* = 2659)	OR (95% CI)	aOR (95% CI)	aOR (95% CI)	aOR (95% CI)
Sex, male *n* (%)	2504 (57.7)	1487 (55.9)	0.93 (0.84–1.03)	0.96 (0.87–1.07)	0.93 (0.84–1.03)	0.94 (0.84–1.04)
Birth weight (100 g), mean (SD)	24.6 ± 8.6	21.5 ± 7.7	0.63 ** (0.59–0.67)	0.74 ** (0.66–0.82)	.	0.81 ** (0.73–0.90)
Birth weight categories (g):						
<1000	228 (5.3)	66 (2.5)	0.46 ** (0.35–0.61)	.	0.14 ** (0.10–0.20)	.
1000–<1500	395 (9.1)	445 (16.7)	2.01 ** (1.74–2.32)		0.78 ** (0.65–0.94)	
1500–<2000	737 (17.0)	804 (30.2)	1		1	
≥2000	2982 (68.7)	1344 (50.5)	0.47 ** (0.42–0.52)		0.67 ** (0.58–0.77)	
Gestational age at birth (w), mean (SD)	35.6 ± 3.9	34.3 ± 3.3	0.91 ** (0.90–0.92)	0.94 ** (0.92–0.96)	0.87 ** (0.85–0.89)	
Gestational age categories (week):						
<28	209 (4.8)	55 (2.1)	0.42 ** (0.31–0.56)			0.26 ** (0.18–0.36)
28–<32	438 (10.1)	458 (17.2)	1.86 ** (1.61–2.14)			0.97 (0.81–1.16)
32–<34	583 (13.4)	650 (24.4)	1			1
34–<37	917 (21.1)	770 (29.0)	1.52 ** (1.36–1.70)			0.71 ** (0.61–0.83)
≥37	2195 (50.6)	726 (27.3)	0.37 ** (0.33–0.41)			0.35 ** (0.29–0.43)
Mothers from other provinces, *n* (%)	2558 (58.9)	1588 (59.7)	1.03 (0.94–1.14)	1.00 (0.88–1.09)	0.98 (0.88–1.09)	0.97 (0.87–1.08)
Cesarean births, *n* (%)	2524 (58.1)	1942 (73.0)	1.95 ** (1.76–2.17)	2.42 ** (2.17–2.71)	2.37 ** (2.12–2.66)	2.24 ** (2.00–2.51)
Born before 2020, *n* (%)	2225 (51.2)	1537 (57.8)	1.30 ** (1.18–1.44)	1.41 ** (1.28–1.56)	1.40 ** (1.26–1.55)	1.37 ** (1.24–1.52)

** *p* < 0.01.

**Table 5 nutrients-16-04402-t005:** Factors associated with a prolonged duration of DHM use in postnatal wards (*n* = 18,419).

	Duration of Using DHM		Binary Analysis	Adjusted Model 1	Adjusted Model 2	Adjusted Model 3	Adjusted Model 4
	Less than 2 days (*n* = 8645)	2 days or more (*n* = 9774)	OR (95% CI)	aOR (95% CI)	aOR (95% CI)	aOR (95% CI)	aOR (95% CI)
Sex, male, *n* (%)	4565 (52.8)	5344 (54.7)	1.08 * (1.02–1.14)	1.05 ^NS^ (0.99–1.12)	1.06 * (1.00–1.13)	1.06 ^NS^ (1.00–1.12)	1.07 * (1.00–1.13)
Birth weight (100 g), mean (SD)	31.3 ± 4.6	31.4 ± 5.0	1.004 (0.999–1.010)	1.0081 * (1.001–1.016)	.	1.08 * (1.00–1.15)	
Birth weight categories (g):							
<2500	685 (7.9)	938 (9.6)	1.23 ** (1.11–1.37)		1.12 ^NS^ (0.99–1.27)	.	
≥2500	7960 (92.1)	8836 (90.4)	1		1	.	
Gestational age at birth (week), mean (SD)	38.4 ± 1.3	38.3 ± 1.3	0.95 ** (0.93–0.97)	0.94 ** (0.92–0.97)	0.97 * (0.95–1.00)	.	
Gestational age categories (week):							
<37	709 (8.2)	1000 (10.2)	1.28 ** (1.15–1.41)			1.29 ** (1.15–1.45)	
≥37	7936 (91.8)	8774 (89.8)	1			1	
Gestational age and birth weight categories:							
LBW–preterm (<2500 g and <37 weeks)	384 (4.4)	543 (5.6)	1.27 ** (1.11–1.45)				1.25 ** (1.09–1.43)
LBW–term (<2500 g and ≥37 weeks)	301 (3.5)	395 (4.0)	1.17 ^NS^ (1.00–1.36)				1.17 ^NS^ (1.00–1.36)
Normal weight–preterm (≥2500 g and <37 weeks)	325 (3.8)	457 (4.7)	1.26 ** (1.09–1.45)				1.21 * (1.04–1.40)
Normal weight-term (≥2500 g and ≥37 weeks)	7635 (88.3)	8379 (85.7)	1				1
Mothers from other provinces, *n* (%)	3752 (43.4)	4155 (42.5)	0.96 (0.91–1.02)	0.90 ** (0.85–0.95)	0.89 ** (0.84–0.95)	0.90 ** (0.84–0.95)	0.89 ** (0.84–0.95)
Cesarean births, *n* (%)	5909 (68.4)	7424 (76.0)	1.46 ** (1.37–1.56)	1.42 ** (1.33–1.52)	1.43 ** (1.34–1.53)	1.43 ** (1.34–1.53)	1.44 ** (1.34–1.54)
Born before 2020, *n* (%)	2585 (29.9)	4787 (49.0)	2.25 ** (2.12–2.39)	2.26 ** (2.12–2.40)	2.26 ** (2.12–2.40)	2.25 ** (2.12–2.39)	2.25 ** (2.12–2.40)

** *p* < 0.01; * *p* < 0.05 but ≥0.01; ^NS^
*p* < 0.10 but ≥0.05

## Data Availability

Requests for data may be directed to the corresponding author and are subject to institutional data use agreements.

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
