# Peer review of "Factors Associated with the Prolonged Use of Donor Human Milk at the Da Nang Hospital for Women and Children in Vietnam"

_nutrients, 2024, doi:10.3390/nu16244402_

Round 1

Reviewer 1 Report

Comments and Suggestions for Authors

This study demonstrates in a very large cohort over several years what supplementation strategies with donated breastmilk (BM) are like and what obstetric factors can influence the maintenance of this nutritional strategy, such as birth weight, route of delivery or prematurity. Some comments to improve the interpretation of the results:

- The abstract must be deeply summarized.

- Introduction. A comprehensible section, however, some sections, such as the relationship of BM with neonatal outcomes, are not the objective of the article, so they could be outside the objective of the work.

- The material and methods are easy to understand and read.

- Results. Are there differences in the variables throughout the analysis time? Figure 1 requires further explanation for its understanding. In table 3, the intake volume is ml/kg/day? It would be more affordable to use that variable. 

- What does it mean that DHM showed an association in a parable-shaping? Could the distribution of points of the regression models be shown in some plot? There is no data on neonatal outcomes at discharge?

Author Response

This study demonstrates in a very large cohort over several years what supplementation strategies with donated breastmilk (BM) are like and what obstetric factors can influence the maintenance of this nutritional strategy, such as birth weight, route of delivery or prematurity. Some comments to improve the interpretation of the results:

Response: Thank you very much 

- The abstract must be deeply summarized.

Response: We have shortened the abstract. 

- Introduction. A comprehensible section, however, some sections, such as the relationship of BM with neonatal outcomes, are not the objective of the article, so they could be outside the objective of the work.

Response: We have shortened the text as suggested.

- The material and methods are easy to understand and read.

Response: Thank you. 

- Results. 
Are there differences in the variables throughout the analysis time? 

Response: There is no difference; we emphasize this in the study data section of the Methods.

Figure 1 requires further explanation for its understanding. 

Response: Explanation in the figure title to make it clearer. 

In table 3, the intake volume is ml/kg/day? It would be more affordable to use that variable. 

Response: In Table 3, we presented total volume (mL) per child. We clarified this in the table title and the text.

- What does it mean that DHM showed an association in a parable-shaping? Could the distribution of points of the regression models be shown in some plot? 

Response: We deleted the term “parable-shaping.” Additionally, we created Figure 2 to highlight some key aORs from the table, as suggested.

There is no data on neonatal outcomes at discharge?

Response: In this study, we were unable to link the information with outcomes. This required linking the data on the use of DHM with medical records, which were mostly paper-based. This will require further work. We acknowledge this limitation.

Reviewer 2 Report

Comments and Suggestions for Authors

I applaud the success of this practice, which provides valuable information that can benefit donor breast milk banks globally. The arbitrary choice of 2020 to separate the retrospective study into two epochs might not be ideal, but it seems reasonable. However, I request some clarifications.

1.    It is hard for me to understand the difference between the neonatal unit and the postnatal ward. Is the neonatal unit equivalent to the level 3/4 unit, or neonatal intensive care unit? Is the postnatal ward equivalent to the level 1-2 nursery? A description of each one is essential for us to interpret the findings.

2.    What is the dependent variable for your logistic regression? Is it the prolonged DBM use?

3.    The study has four logistic regressions. Do you include all subjects in the first three regressions or those handled in the neonatal unit?

4.    I do not understand why you describe “in the first six years of operations” in the first sentence of the results. First, your HMB was established in 2017, so “the first six years” make no sense. Secondly, do you have more than one operation?

5.    The 70% Cesarean section rate is amazingly high for those who received DBM. It is well known that C/S affects the success of breast milk feeding due to concerns, or parent’s perceptions, about the possible negative influence of anesthetics on babies. Is the C/S rate the same for those not received DBM?

6.    Are there guidelines to dictate the priority of who can receive DBM? Many facilities are facing the problem of not having enough DBM for all hospitalized neonates.

7.    In Table 2, there is a (1.3-5) for 2017 <1000 g group. Should it be 1.0? I am confused that the median duration of using DBM for postnatal wards ranges between 1-24, with 24 never appearing in any year.

8.    Figure 1 provides part of the data in Table 1 which becomes redundant.

9.    I will suggest you use a U-shape instead of a parabola shape so that average readers can understand.

10.  Please rearrange Table 5. The arrangement makes it very difficult to interpret.

11.  incredibly, all premature neonates require only a very short duration of DBM, were they all successfully transition to exclusive BM from their mothers? Did some of them transition to premature formula?

Author Response

I applaud the success of this practice, which provides valuable information that can benefit donor breast milk banks globally. The arbitrary choice of 2020 to separate the retrospective study into two epochs might not be ideal, but it seems reasonable. However, I request some clarifications.

Response: Thank you very much 

1.    It is hard for me to understand the difference between the neonatal unit and the postnatal ward. Is the neonatal unit equivalent to the level 3/4 unit, or neonatal intensive care unit? Is the postnatal ward equivalent to the level 1-2 nursery? A description of each one is essential for us to interpret the findings.

Response: We make it clearer in the text under Study setting of the Methods. 

2.    What is the dependent variable for your logistic regression? Is it the prolonged DHM use?

Response: Yes, it is. We called it outcome variable. We changed it to "dependent, outcome variable” to make it clearer. 

3.    The study has four logistic regressions. Do you include all subjects in the first three regressions or those handled in the neonatal unit?

Response: We have three logistic regression models for newborns received DHM at neonatal units (n=7100). Then we have four logistic regression models for newborns received DHM at postnatal wards (n=18419). We have included the sample size in the title for tables 4 and 5 to make it clearer.

4.    I do not understand why you describe “in the first six years of operations” in the first sentence of the results. First, your HMB was established in 2017, so “the first six years” make no sense. Secondly, do you have more than one operation?

Response: We change to operation (singular) as suggested.  We also spell out the duration instead of first six years of operation.

5.    The 70% Cesarean section rate is amazingly high for those who received DBM. It is well known that C/S affects the success of breast milk feeding due to concerns, or parent’s perceptions, about the possible negative influence of anesthetics on babies. Is the C/S rate the same for those not received DBM?

Response: 

We discussed about this in the discussion section “Cesarean sections as a barrier to breastfeeding and DHM use” and acknowledge as our limitation.

6.    Are there guidelines to dictate the priority of who can receive DBM? Many facilities are facing the problem of not having enough DBM for all hospitalized neonates.

This hospital and Vietnam have HMB guidelines regarding the priority of prescribing DHM. We described this information under the study setting in the Methods section and cited the relevant policies.

7.    In Table 2, there is a (1.3-5) for 2017 <1000 g group. Should it be 1.0? I am confused that the median duration of using DBM for postnatal wards ranges between 1-24, with 24 never appearing in any year.

Response: We corrected those typos. Additionally, it should be 14, not 24. Thank you.

8.    Figure 1 provides part of the data in Table 1 which becomes redundant.

Response: Indeed, the information are different. Figure 1 is on the distribution (%) of duration of using DHM (days) in neonatal units (A) and postnatal wards (B) by year; while in Table 1, we presented characteristics of infants receiving DHM at The Da Nang Hospital for Women and Children by year. We revised the Figure title to make it clearer. 

9.    I will suggest you use a U-shape instead of a parabola shape so that average readers can understand.

Response: We deleted all mentions of “parabola-shape” in the manuscript.

10.  Please rearrange Table 5. The arrangement makes it very difficult to interpret.

Response: We rearranged Tables 4 and 5 to make them easier to follow. Additionally, we created Figure 2 to highlight some key aORs from the table, as suggested.

11.  incredibly, all premature neonates require only a very short duration of DBM, were they all successfully transition to exclusive BM from their mothers? Did some of them transition to premature formula?

Response: his is a formula-free hospital. Thus, all infants transition to exclusive breastmilk from their mothers. We clarified this in the study variables section of the Methods.